# A leap in quantum efficiency through light harvesting in photoreceptor UVR8

Xiankun Li[1,2], Haisheng Ren [3,4], Mainak Kundu[1], Zheyun Liu[1], Frank W. Zhong[1,5], Lijuan Wang[1], Jiali Gao [3,6✉] & Dongping Zhong [1,2✉]

Plants utilize a UV-B (280 to 315 nm) photoreceptor UVR8 (UV RESISTANCE LOCUS 8) to sense environmental UV levels and regulate gene expression to avoid harmful UV effects. Uniquely, UVR8 uses intrinsic tryptophan for UV-B perception with a homodimer structure containing 26 structural tryptophan residues. However, besides 8 tryptophans at the dimer interface to form two critical pyramid perception centers, the other 18 tryptophans' functional role is unknown. Here, using ultrafast fluorescence spectroscopy, computational methods and extensive mutations, we find that all 18 tryptophans form light-harvesting networks and funnel their excitation energy to the pyramid centers to enhance light-perception efficiency. We determine the timescales of all elementary tryptophan-to-tryptophan energy-transfer steps in picoseconds to nanoseconds, in excellent agreement with quantum computational calculations, and finally reveal a significant leap in light-perception quantum efficiency from 35% to 73%. This photoreceptor is the first system discovered so far, to be best of our knowledge, using natural amino-acid tryptophans to form networks for both light harvesting and light perception.

---

[1] Department of Physics, Department of Chemistry and Biochemistry, Programs of Biophysics, Chemical Physics and Biochemistry, The Ohio State University, Columbus, OH 43210, USA. [2] Center for Ultrafast Science and Technology, School of Physics and Astronomy, School of Chemistry and Chemical Engineering, Shanghai Jiao Tong University, Shanghai 200240, China. [3] Department of Chemistry and Supercomputing Institute, University of Minnesota, Minneapolis, MN 55455, USA. [4] College of Chemical Engineering, Sichuan University, Chengdu 610065, China. [5] Cell and Molecular Biology Program, University of Chicago, Chicago, IL 60637, USA. [6] School of Chemical Biology and Biotechnology, Peking University Shenzhen Graduate School, Shenzhen 518055, China.
✉email: jiali@jialigao.org; zhong.28@osu.edu

UVR8 protein has been identified as a UV-B (280–315 nm) photoreceptor in plants[1]. UVR8 forms a homodimer (Fig. 1a) that dissociates into two monomers in response to UV-B irradiation and thus triggers various UV protective mechanisms via cell signaling processes[1–12]. Followed by initial light perception, monomeric UVR8 interacts with signaling partners including E3 ubiquitin-protein ligase COP1 (CON-STITUTIVE PHOTOMORPHOGENIC1) and accumulates in the nucleus, regulating expression of various downstream genes[1–12]. Unlike previously discovered visible and infrared photo-receptors[13–15], UVR8 does not contain an external chromophore but utilizes the natural amino-acid tryptophan (W or Trp) for light perception[16,17]. Each UVR8 monomer has 14 tryptophan residues, and except the unstructured C-terminal one, the rest 13 structural Trp residues could be classified into three distinct groups, a distal ring (6$W_d$ in Fig. 1b), a peripheral outlier (3$W_p$ in Fig. 1c), and a pyramid center (4$W_c$ in Fig. 1c). Although all tryptophan residues can absorb UV-B radiation, the critical reaction leading to dimer dissociation occurs only in the interfacial pyramid center[1,16–26]. One central question in understanding light perception by UVR8 is the collective roles of all tryptophan residues: is there light harvesting in UVR8 to enhance its quantum yield for sensitive detection of the relatively low intensity of UV-B radiation? We have recently examined the possible energy transfer to the pyramid center[19], which further gained theoretical support[20,21]. However, the entire excitation-energy-transfer processes remain unknown. Here, using ultrafast fluorescence spectroscopy, extensive site-directed mutations, computational methods and Förster resonance energy-transfer theory (RET), we map out complete excitation-energy-transfer network along with all the transfer timescales in UVR8, revealing the mechanism of energy flow from the distal and peripheral tryptophans to the two pyramid perception centers at the dimer interface.

## Results

**Spectra of three tryptophan groups.** To elucidate the collective functions of the three groups of tryptophan residues, we first

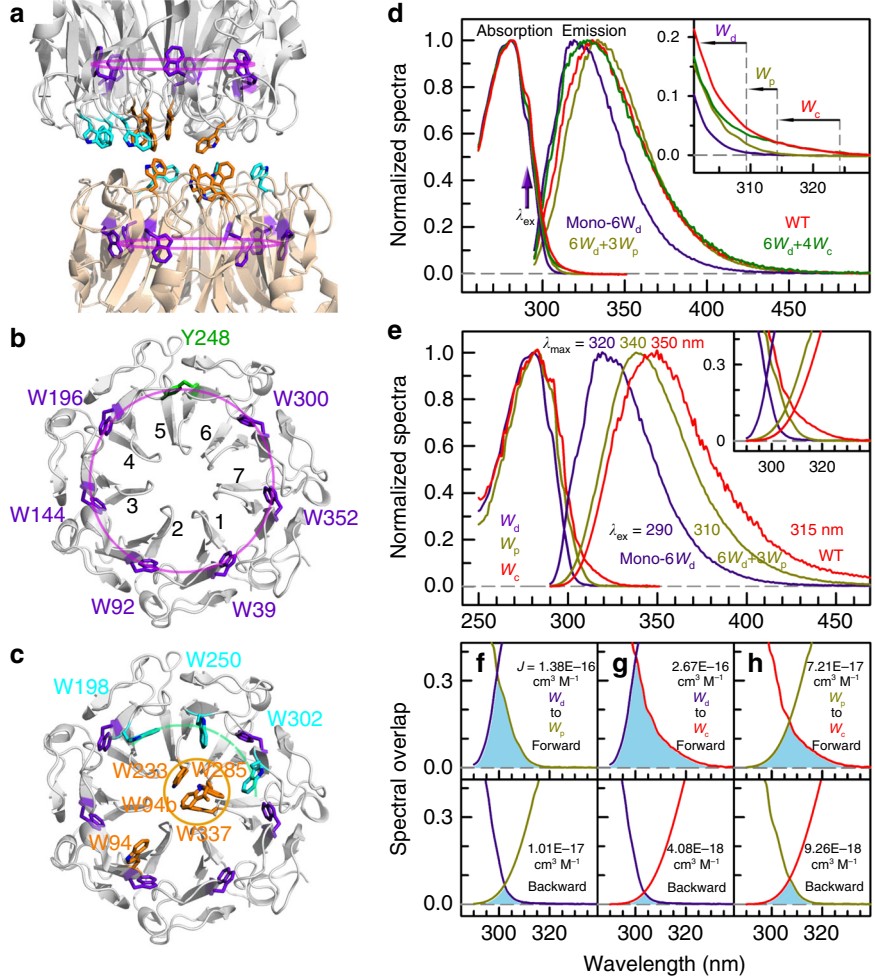

**Fig. 1 UVR8 light-harvesting tryptophan networks and their steady-state spectra. a** A side view of UVR8 dimer with three groups of W residues highlighted in different colors: 6$W_d$, purple; 3$W_p$, cyan; and 4$W_c$, orange. 3$W_p$ and 4$W_c$ lie at the dimer interface whereas 6$W_d$ are buried in the middle of β sheets. **b** A top view of one subunit from the dimer interface shows the highly symmetrical ring locations of 6$W_d$ (W39, W92, W144, W196, W300, and W352). **c** A top view from the dimer interface displays the relative positions of the pyramid center (W337, W285, W233, and W94b) and the 3$W_p$ peripheral outlier (W302, W250, and W198). **d** Normalized absorption and emission spectra for UVR8 WT (red), 6$W_d$ + 4$W_c$ (green), 6$W_d$ + 3$W_p$ (dark yellow), and mono-6$W_d$ (purple). The inset highlights absorption differences beyond 300-nm wavelength. **e** Normalized absorption and emission spectra for 6$W_d$ (purple), 3$W_p$ (dark yellow), and 4$W_c$ (red). The inset shows a close view of the absorption–emission overlap region. Note the different excitation wavelengths for selected W groups; 290 nm for 6$W_d$, 310 for 3$W_p$, and 315 for 4$W_c$. **f–h** Spectral overlap integrals for $W_d$-to-$W_p$ (**f**, top), $W_d$-to-$W_c$ (**g**, top), and $W_p$-to-$W_c$ (**h**, top) energy transfer. The corresponding reverse transfer is shown at the bottom of each panel. The overlap regions are shaded by light blue color. Note that the forward $J$ refers $W_d$ to $W_p$ and $W_c$ or $W_p$ to $W_c$, while the backward $J$ refers $W_p$ to $W_d$, $W_c$ to $W_d$ and $W_p$.

prepared three critical mutants (Supplementary Table 1) using site-directed mutagenesis to obtain their absorption and emission spectra; we mutated all four tryptophan residues in the $4W_c$ pyramid center (denoted as $6W_d + 3W_p$), the entire $3W_p$ peripheral outlier ($6W_d + 4W_c$), and both $W_p$ and $W_c$ groups (mono-$6W_d$), all into phenylalanine residues. The first two mutants can still form dimers, but the latter is a monomer. Distinctively different absorption spectra were observed in the region beyond 300 nm (inset in Fig. 1d). For the WT and the mutant $6W_d + 4W_c$, the absorption spectra extend to above 330 nm and such red-tail lowest energy absorption comes from the closely packed $4W_c$, a signature of possible exciton formation[16]. The red-side absorption of $4W_c$ explains the recently reported UVR8 monomerization in response to UV-A light up to 350-nm wavelength[27]. For the mutant $6W_d + 3W_p$, the absorption is blue-shifted due to the removal of $4W_c$, still extending to 315 nm. The mono-$6W_d$ mutant by deleting all seven interfacial tryptophan residues from each subunit has absorption further blue-shifted to below 310 nm, revealing that the distal $6W_d$ exhibit the highest energy absorption among the three groups. From Fig. 1d, we readily obtained the absorption spectra of the separated three groups of $6W_d$, $3W_p$ and $4W_c$ and their corresponding fluorescence spectra by excitation of the three mutants of mono-$6W_d$, $6W_d + 3W_p$, and $6W_d + 4W_c$ (or WT) at 290, 310, and 315 nm, respectively (Fig. 1e). Importantly, the spectral differences allow us to selectively excite each group of tryptophans to obtain their individual emission spectra. In particular, 315-nm light only excites $4W_c$ in $6W_d + 4W_c$ or the WT, and 310-nm light only excites $3W_p$ in $6W_d + 3W_p$, plus 290-nm excitation of mono-$6W_d$. The three emission spectra have distinct peak maxima at 320 nm for $6W_d$, 340 nm for $3W_p$, and 350 nm for $4W_c$. These steady-state spectra reveal a striking hierarchy of energetic order and direction of excited energy flow. There are clear spectral overlaps between absorption and emission among the three tryptophan groups (inset in Fig. 1e–h), resulting in excitation-energy transfer in one direction from $6W_d$ to $3W_p$ and $4W_c$ and from $3W_p$ to $4W_c$, i.e., the energy flows from the distal ring to the dimer interface and from the interfacial peripheral outlier to the pyramid center. The spectral overlap integrals from Fig. 1f–h for $6W_d$-to-$3W_p$, $6W_d$-to-$4W_c$, and $3W_p$-to-$4W_c$ are $1.38 \times 10^{-16}$, $2.67 \times 10^{-16}$, and $7.21 \times 10^{-17}$ cm$^3$ M$^{-1}$, respectively, and the reverse transfer integrals are one to two orders of magnitude smaller and thus negligible (Fig. 1f–h).

**Energy transfer from $W_d$ to $W_p$ and $W_c$.** We next used time-resolved fluorescence spectroscopy to directly measure the excitation-energy-transfer processes from $6W_d$. We precisely measured the lifetimes of $6W_d$ (mono-$6W_d$) without the energy acceptors and obtained two lifetimes of 500 ps (29% amplitude) and 2.7 ns (71%) (Fig. 2a). Then, we designed a sequence of mutants of $6W_d$ with gradual introduction of tryptophan residues at the dimer interface: $1W_p$ (Fig. 2b and Supplementary Fig. 2a, b), $2W_p$ (Fig. 2c and Supplementary Fig. 2c, d), $3W_p$ (Fig. 2d), and $4W_c$ (Fig. 2e). For each mutant, we took 16 fluorescence decay transients gated from the blue to red-side emission wavelengths to determine the lifetime changes of $6W_d$ in the presence of these acceptors (Supplementary Fig. 1). As shown in Fig. 2b–f, we constructed the lifetime-associated spectra for all observed decay lifetimes from a global fitting of the fluorescence transients. For the 7 possible combinations of mutants containing $W_p$ (Fig. 2b–d and Supplementary Fig. 2a–d), we observed three timescales in 0.5, 1.9–2.4, and 6–8 ns. The 6–8 ns components must come from the $W_p$ contributions since this lifetime was not observed in mono-$6W_d$ and their associated spectra (squares) have the same shape as the directly measured emission spectra (solid lines) by

310-nm excitation. The 0.5 ns and 1.9–2.4 ns components are from the $6W_d$ emission in the presence of acceptors. The lifetime-associated spectra (circles and triangles) are nearly the same as the directly resolved emission spectra (solid lines) from mono-$6W_d$ (Supplementary Fig. 3). Based on the lifetime change from 2.7 to 1.9–2.4 ns, we thus derived the total energy-transfer times in 6.9–23 ns for various $W_p$ mutants (Supplementary Table 2). For the 0.5-ns lifetime component, such slow energy transfer in 6.9–23 ns will not make noticeable changes.

For the mutant $6W_d + 4W_c$ (Fig. 2e) and WT (Fig. 2f), the 16 transients are best fit globally with 4 time constants. Since the resonance energy-transfer (RET) rates from $6W_d$ to $4W_c$ are expected to be faster based on the protein structure and theoretical estimates (see below), the 0.5 and 2.7 ns components of $6W_d$ (Fig. 2a) becomes 0.45 and 1.2–1.5 ns for $6W_d + 4W_c$ and WT. Thus, we determined total energy-transfer efficiency of 45% from the interior $6W_d$ to the interfacial tryptophan residues $3W_p$ and $4W_c$. The two additional times of 1.4 and 5.4 ns for $6W_d + 4W_c$ are from $4W_c$ (Supplementary Fig. 4) but the 1.4 and 5.5 ns for the WT are from $4W_c$ and $3W_p$, respectively. For the former, the resulting emission spectrum (hexagons in Fig. 2e) is the same as directly obtained by excitation at 315 nm (solid line). For the latter, the spectrum (hexagons in Fig. 2f) can be readily decomposed into the directly measured $4W_c$ and $3W_p$ spectra (dashed lines).

We then used a combined quantum mechanical and molecular mechanical (QM/MM) method to determine the tryptophan transition dipole moments using 500 snapshots (4-ns interval) from a 2-μs MD simulation, and applied the FRET theory[28,29] to obtain excitation-energy-transfer rate distributions for all possible tryptophan donor–acceptor pairs (Supplementary Fig. 5). A total number of 6500 (500 snapshots × 13 tryptophan) transition dipole moments were calculated and 42,000 (500 × 6 × 14) transfer rates for the dimer were calculated to generate 12 (6 × 2) rate distributions of each $W_d$ to $3W_p$ and $4W_c$ (Fig. 2g), revealing a clear preference for energy transfer to the pyramid center, mostly to W285 and W233 (Supplementary Table 3). In Fig. 2h–j, we simulated the $6W_d$ decay dynamics of the original 2.7-ns component with the calculated RET rate distributions. Significantly, even though each $W_d$ donor has different decay dynamics, due to various RET rates, the total $6W_d$ dynamics can be perfectly described with a single-exponential decay and the time constant is in excellent agreement with the experimental value in all mutants (Fig. 2h–j and Supplementary Fig. 2e–j), demonstrating that our proposed scheme is accurate for energy transfer from $W_d$ to $W_p$ and $W_c$. As summarized in Fig. 2k, from the $6W_d$ ring network to $3W_p$ and $4W_c$, each $W_d$ has 14 energy-transfer pathways and there are in total 168 energy-transfer pairs in the dimer. Notably, the distal $W_d$ from one subunit can donate excitation energy to the interfacial $W_p$ and $W_c$ from both subunits. Although the energy-transfer rates have broad distributions due to protein fluctuations (Supplementary Fig. 5), we can obtain effective RET time constants ($\tau_{RET}$) for each transfer pair from numerical simulations (Supplementary Fig. 6 and Supplementary Table 3). By summing the effective rate constants of all parallel energy-transfer pathways for each $W_d$, the total RET time constants ($\tau_{total}$ in Fig. 2k) are within a few nanoseconds, comparable to their lifetimes. Collectively, our model suggests that $6W_d$ donate excitation energy to the 14 interfacial tryptophan residues (7 from each monomer) with total transfer efficiency of 44% in WT (see energy-transfer efficiency calculations in Methods and Supplementary Table 4), in excellent accord with the experimental result (~45%) observed above.

**Energy transfer from $W_p$ to $W_c$.** We next investigated the energy flow from the $3W_p$ peripheral outlier to $4W_c$; each $W_p$ has four

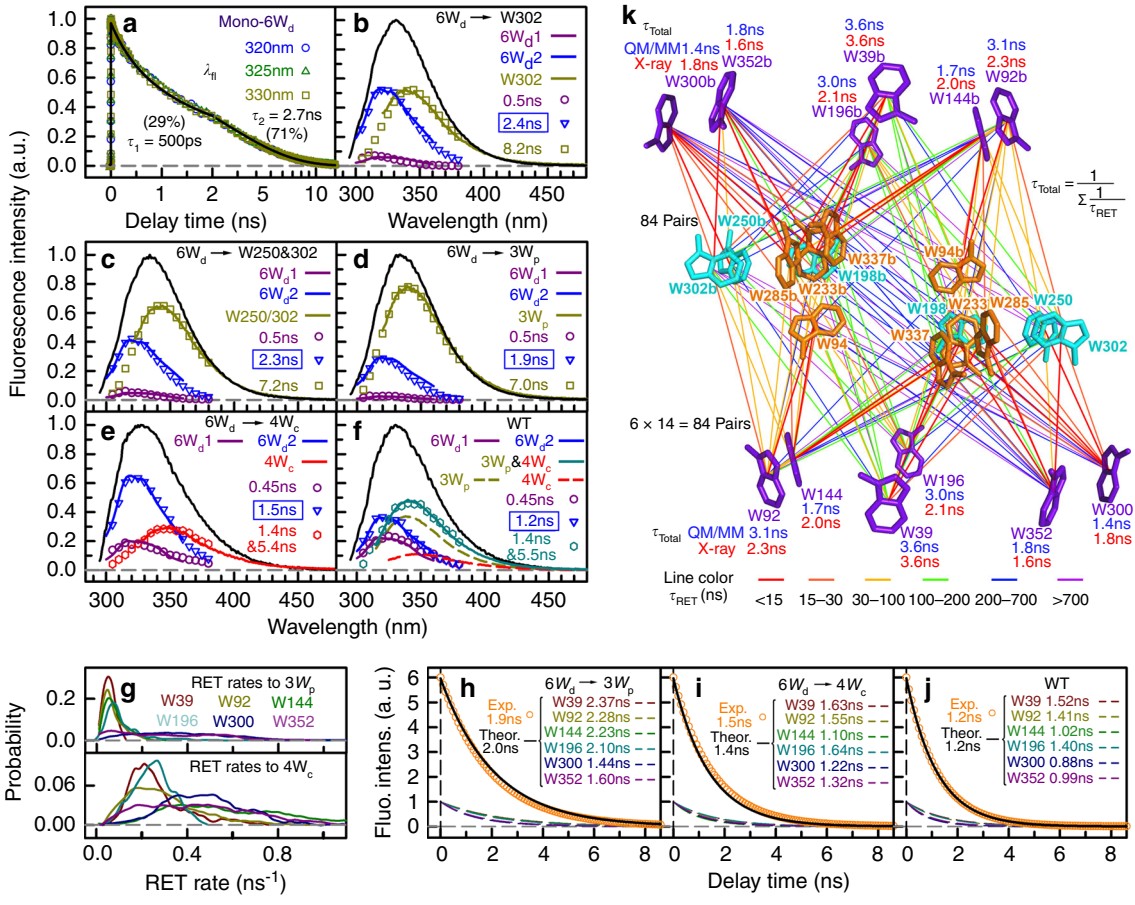

**Fig. 2 Excitation-energy transfer from $W_d$ to $W_p$ and $W_c$ in UVR8. a** Fluorescence transients for mono-$6W_d$ gated at 320, 325, and 330 nm. Note the log scale starting from 2 ns. The lifetimes of $W_d$ were determined to be $\tau_1 = 0.5$ ns (29%) and $\tau_2 = 2.7$ ns (71%). **b–f** Timescales from a global fitting and lifetime-associated spectra of selected $6W_d + 1W_p$ (**b**), $6W_d + 2W_p$ (**c**), $6W_d + 3W_p$ (**d**), $6W_d + 4W_c$ (**e**), and WT (**f**). Fluorescence transients are shown in Supplementary Fig. 1. For each panel, the solid lines are the steady-state emission spectra of various mutants. The lifetime-associated spectra are shown in various symbols and the directly measured spectra are shown in lines. The time constant corresponding to the original 2.7-ns component is highlighted in a box. $6W_d1$ and $6W_d2$ are the emission spectra of the two $W_d$ lifetimes decomposed as described in Methods. Note that the total spectrum of 1.4 and 5.4 ns shown for $6W_d + 4W_c$ agrees with the $4W_c$ emission. The total spectrum of 1.4 and 5.5 ns shown for WT can be decomposed into $3W_p$ and $4W_c$ emission (dashed lines). **g** Energy-transfer rate distributions for each $W_d$ to $3W_p$ (top) and to $4W_c$ (bottom) based on QM/MM methods. **h–j** Simulations of the original 2.7-ns component decay dynamics of $6W_d$ for $6W_d + 3W_p$ (**h**), $6W_d + 4W_c$ (**i**), and WT (**j**) based on RET rate distributions. For each case, the overall $6W_d$ decay curve (black line), the sum of six individual $W_d$ decay curves (dashed lines), can be fitted with a single-exponential decay and agrees well with the experimental decay dynamics shown in (**d–f**). **k** The 168 possible energy-transfer pathways (based on QM/MM) from all $W_d$ to 14 interfacial W residues ($3W_p + 4W_c$ on both subunits). The total RET time constants based on QM/MM (blue) and on X-ray structure (red) are shown near each $W_d$. Each line represents one energy-transfer pathway. Colors of the lines are based on effective RET time constants. The dominant paths are shown in red.

dominant energy-transfer pathways. Depicted in Fig. 3a–c are rate distributions of the 12 energy-transfer pairs from QM/MM calculations, showing again that the dominant transfer is to W285 and W233 in the pyramid center. Figure 3d shows that the energy transfer from each $W_p$ to $4W_c$ can be effectively described with a fast transfer ($\tau_{total1}$) and a slow transfer ($\tau_{total2}$) rate, resulting from structural fluctuations of UVR8. W302 and W250 have ultrafast transfer times in 80 and 120 ps ($\tau_{total1}$), respectively, while W198 has a transfer time of 1.8 ns owing to unfavorable orientations and longer distances. Figure 3e summarizes all light-harvesting tryptophan networks from $3W_p$ to $4W_c$ at the dimer interface.

Experimentally, we separately measured each $W_p$ lifetime because each $W_p$ has different local environment, using the 310-nm excitation for each $W_p$ to avoid $W_d$ absorption. In Fig. 3f and Supplementary Fig. 7, we determined the lifetimes of peripheral W302, W250, and W198 one by one using $6W_d + 1W_p$ mutants. The dominant lifetime is 6–8 ns. Furthermore, the fluorescence decay transients of the $6W_d + 2W_p$ and $6W_d + 3W_p$ mutants are

in good agreement with the sum of the individual $W_p$ decay dynamics measured in $6W_d + 1W_p$ mutants (Supplementary Fig. 8), indicating negligible energy transfer among $3W_p$. We also measured the fluorescence dynamics of $4W_c$ in WT by only exciting the pyramid center at 315 nm and Fig. 3g shows the fluorescence transients in two timescales, 80 ps (75%) and 1.4 ns (25%), a result of the structural fluctuations of the pyramid center at the interface. The transients at 340 nm among various fluorescence wavelengths (Supplementary Figs. 9–12) are shown in Fig. 3h–k, for selected $W_p$ mutants. With the total RET rates and their ratios (Fig. 3d), and the known lifetimes above, we simulated all transients; they are completely consistent with the experimental data (Fig. 3h–k, and Supplementary Figs. 9–12) and the resulting spectra also are in good accord with the directly measured ones (Fig. 3l–n). Finally, we obtained transfer efficiency of 96% for W302, 94% for W250, and 63% for W198 to $4W_c$, and an overall average transfer efficiency of 84% from $3W_p$ to $4W_c$ (see energy-transfer efficiency calculations in Methods and Supplementary Table 4). Combining all above data, we simulated

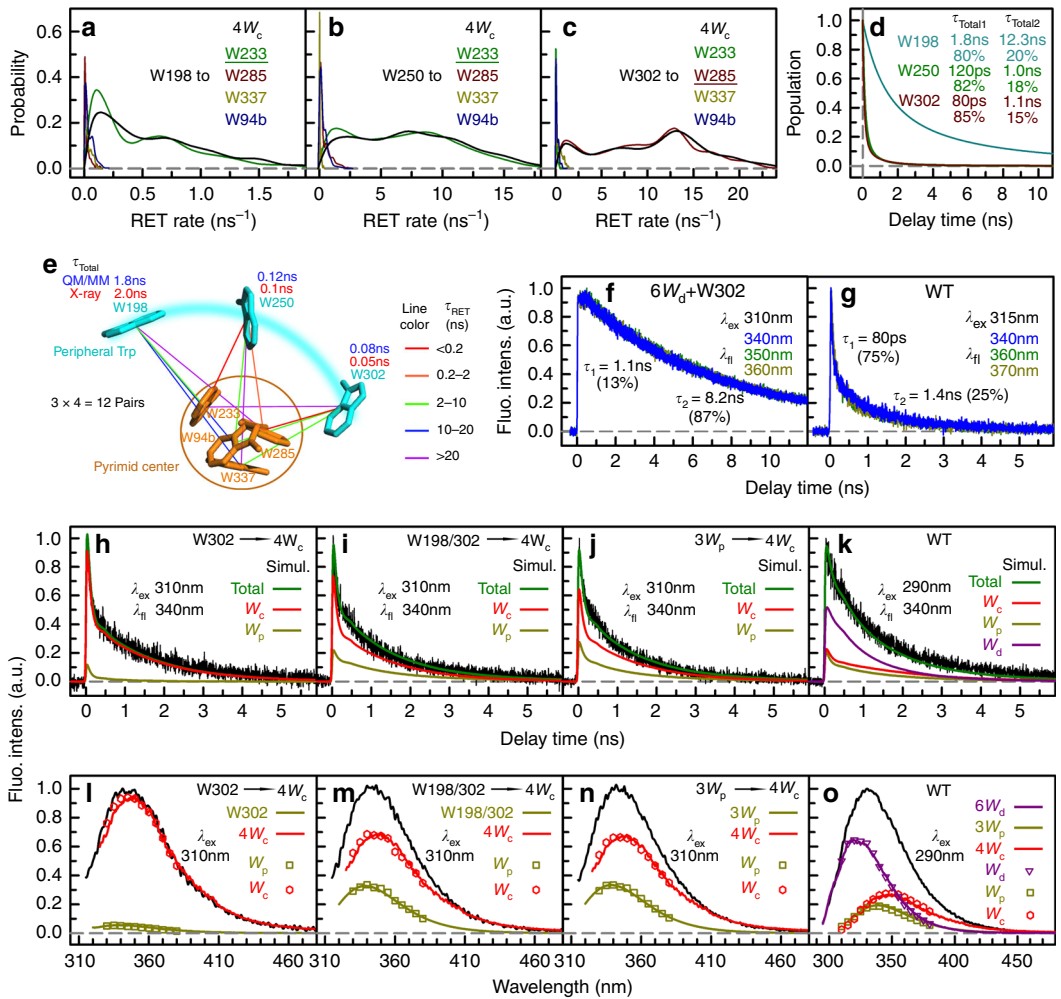

**Fig. 3 Excitation-energy transfer from $W_p$ to $W_c$. a–c** RET rate distributions based on QM/MM calculations for each $W_p$–$W_c$ pair. **a** W198 as the donor.
**b** W250 as the donor. **c** W302 as the donor. In each figure, the distribution of the total RET rate to $4W_c$ is shown in black and the most dominant energy
acceptor is underscored. **d** Simulation of energy-transfer dynamics for $3W_p$ to $4W_c$ based on RET rate distributions. For all $3W_p$, a fast energy transfer
($\tau_{total1}$) and a slow energy transfer ($\tau_{total2}$) are needed to fit the transients. **e** The 12 possible energy-transfer pathways from $3W_p$ to $4W_c$ as calculated with
the FRET theory. Each line represents one energy-transfer pathway. Colors of the lines are based on calculated RET time constants. The total RET time
constants are shown near each $W_p$. **f** Fluorescence dynamics of selected $6W_d + 1W_p$ at excitation of 310 nm. **g** Fluorescence dynamics of WT at excitation
of 315 nm. **h–k** Model simulations of typical fluorescence transients at 340 nm for mutants $6W_d + 1W_p + 4W_c$(**h**), $6W_d + 2W_p + 4W_c$ (**i**), $6W_d + 3W_p +$
$4W_c$ (**j**), and WT (**k**). In each figure, the black solid line is the experimental data and the green solid line is the total simulation curve, which is the sum of
contributed ones from $W_c$ (red line), $W_p$ (dark yellow line), and $W_d$ (purple line). The simulation method is detailed in Methods and Supplementary
Methods. **l–o** Comparison between directly measured and simulation-constructed spectra for $6W_d + 1W_p + 4W_c$ (**l**), $6W_d + 2W_p + 4W_c$ (**m**), $6W_d +$
$3W_p + 4W_c$ (**n**), and WT (**o**). In each panel, the black line is the total emission. The colored symbols represent the spectra of three W groups decomposed
from the total emission based on the time integrals of simulation curves (Supplementary Figs. 9–13). Solid lines are directly measured emission spectra of
the three W groups.

the fluorescence transients for the WT UVR8 at excitation of 290
nm with all three W groups included and the results are in good
agreement with experimental data (Fig. 3k and Supplementary
Fig. 13). The constructed spectra corresponding to the three W
groups are similar to those measured directly (symbols vs. solid
lines in Fig. 3o), further validating our proposed excitation-
transfer model.

## Discussion
To recapitulate, based on the absorption coefficients of three W
groups and the transfer mechanism presented in Fig. 4, for every
100 photons absorbed at 290 nm, 40 photons excite $W_d$, 25 excite
$W_p$, and 35 excite $W_c$ (see tryptophan absorbance ratios in
Supplementary Table 5). Among the excited 40 $W_d$, about 12.4

$(40 \times 31\%)$ are transferred to $W_c$ and 5 $(40 \times 13\%)$ are transferred
to $W_p$. The total excited $W_p$ are 30 $(25 + 5)$ and 25.2 $(30 \times 84\%)$
are transferred to $W_c$. Thus, the final excited $W_c$ population is 73
out of 100; 35 are directly excited and 38 are transferred, leading
to significantly enhanced light-perception efficiency from 35%
(direct excitation) to 73% (light harvesting). Clearly, besides the
possible structural roles of those tryptophans, the distal and
peripheral tryptophan networks do play a functional role to
harvest and funnel UV-B energy into the pyramid perception
centers (Fig. 4) to ignite the reaction and unzip the dimer
interface for signaling. UVR8 is the first system observed so far, to
be best of our knowledge, to utilize intrinsic amino acids (tryp-
tophan) to form the light-harvesting networks as well as to act
as light-perception receptors. Since the tryptophan network is
conserved among various species[30–32], the light-harvesting

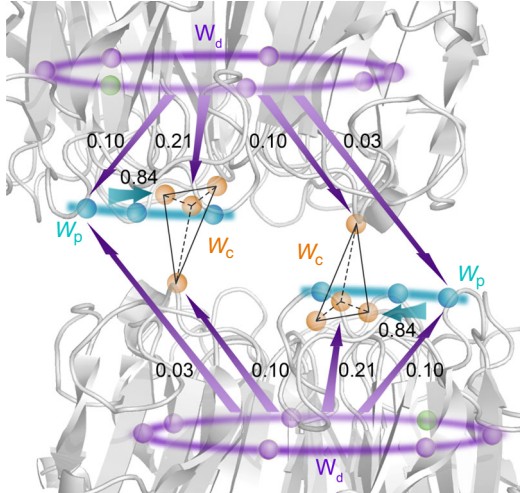

**Fig. 4 A scheme of excitation-energy-transfer networks in UVR8.** The two monomers are shown in gray. $6W_d$, $3W_p$, $4W_c$, and Y248 are shown in purple, cyan, orange, and green spheres, respectively. Arrows show excitation-energy flow directions from the distal ring to the interfacial peripheral outliers and pyramid centers, and from the peripheral outlier to the pyramid center. The corresponding values are the branching fractions for their energy-transfer pathways.

mechanism likely has arisen early in the evolution of UVR8 photoreceptors. Clearly, the beautiful architecture of these tryptophan residues in three groups is not only for the structure integrity but also more importantly for efficient light harvesting and subsequent chemical reactions for initiating biological function.

## Methods

**UVR8 protein sample preparation.** *Arabidopsis thaliana* UVR8 protein samples for this study were prepared following the protocol reported by Wu et al.[17]. Plasmids expressing UVR8 mutants were prepared with Quickchange site-directed mutagenesis kits (Qiagen) following the manufacturer's protocol. For all experiments, protein samples were kept in the UVR8 lysis buffer (150 mM NaCl, 25 mM Tris, and pH = 8.0) unless stated otherwise. Knockout of all the 3 peripheral and 4 pyramid center tryptophan residues does not yield a well-folded protein. However, a well-folded protein can be obtained when an additional residue R286 is mutated (W285/233/94/337/198/250F/W302H/R286A or mono-$6W_d$), which makes the mutant monomeric even without UV-B. The C-terminus (amino acid 390–440) of *Arabidopsis thaliana* UVR8 is intrinsically unstructured and does not play a role in UVR8 light-induced dissociation[16,17]. Thus, the C-terminal tryptophan W400 was mutated to phenylalanine (F) in all UVR8 proteins in our studies. For simplicity, we redefine W400F as the wild type (WT) throughout our paper and all site-directed mutations were based on this template. All UVR8 mutants used in this study were summarized in Supplementary Table 1. All experiments were conducted at room temperature.

**Steady-state absorption and emission spectra.** The absorption spectra were measured using Cary 50 UV-Vis Spectrophotometer (Agilent), with the OD at absorption peak (282 nm) adjusted to around 1.0 in 5-mm quartz cuvettes for all samples (20–50 μM dimer). Absorption spectra of the three W groups were obtained with the normalized absorption spectra of UVR8 WT, $6W_d + 3W_p$, and mono-$6W_d$ using the following equations:

$$A_d(\lambda) = A_{\text{mono}-6W_d}(\lambda) \tag{1}$$

$$A_p(\lambda) = \frac{9 \times A_{6W_d+3W_p}(\lambda) - 6 \times A_{\text{mono}-6W_d}(\lambda)}{3} \tag{2}$$

$$A_c(\lambda) = \frac{13 \times A_{\text{WT}}(\lambda) - 9 \times A_{6W_d+3W_p}(\lambda)}{4} \tag{3}$$

Where $A_{\text{mono}-6Wd}$, $A_{6Wd+3Wp}$, and $A_{\text{WT}}$ are normalized absorption spectra of UVR8 mono-$6W_d$, $6W_d + 3W_p$, and WT (as shown in Fig. 1d), respectively. $A_d$, $A_p$, and $A_c$ are the obtained normalized absorption spectra of distal, peripheral, and pyramid center tryptophan residues (as shown in Fig. 1e), respectively.

The emission spectra of UVR8 samples were measured in 5-mm quartz cuvettes using Fluoromax-3 spectrofluorimeter (Horiba) at UVR8 dimer concentrations of about 2, 10, 50, and 50 μM, for excitation wavelengths at 290, 300, 310, and 315 nm, respectively.

**Log-normal fitting of tryptophan fluorescence spectra.** $W_p$ and $W_c$ emission spectra with wavelengths close to or shorter than the excitation wavelength were extended by fitting with four-parameter log-normal functions[33]. The procedure was detailed elsewhere[33]. The results are shown in Supplementary Fig. 14.

**Calculations of FRET spectral overlap integral J.** Details about FRET theory can be found elsewhere[28], J value was calculated as follows:

$$J = \frac{\int_0^\infty F_D(\lambda)\varepsilon_A(\lambda)\lambda^4 d\lambda}{\int_0^\infty F_D(\lambda)d\lambda}, \tag{4}$$

where $F_D$ is the normalized emission spectrum of the donor ($W_d$ or $W_p$). The area of $F_D$ was normalized to 1. $\varepsilon_A(\lambda)$ is the extinction coefficient of the acceptor (in $M^{-1}$ $cm^{-1}$). $\varepsilon_A(\lambda)$ has the same shape as $W_p$ or $W_c$ absorption spectra with the maxima adjusted to 5600 $M^{-1}$ $cm^{-1}$, which is the literature molar extinction coefficient of tryptophan in proteins at the absorption peak[34].

**Fluorescence quantum yield (QY) determination.** Fluorescence spectra of samples with a series of concentrations were measured with 290/310/315 nm excitation and then integrated, and the integrals were then plotted versus absorbance at 290/310/315 nm. For 290-nm excitation, tryptophan in water was used as a standard[35]. For 310- and 315-nm excitation, phenanthrene in absolute ethanol (under anaerobic condition) was used as a standard[36]. For each sample, three independent measurements were conducted, and the error bars were determined using the standard deviation of the three measurements. The results are shown in Supplementary Fig. 15.

**Femtosecond-resolved fluorescence up-conversion method.** The entire experimental layout has been detailed elsewhere[37]. Briefly, the pump wavelength was set at 290 nm, generated by OPA-800 (1 kHz, Specta-Physics), and its pulse energy was attenuated to 100 nJ. The instrument response time under the current noncollinear geometry is about 300 fs, and all data were taken at a magic angle (54.7°). The maximum time window is about 3 ns. The samples (150 μM) were kept in spinning quartz cells during irradiation to avoid heating and photobleaching. Samples in the cells were replaced with fresh samples about every 1 million pump pulses.

**Picosecond-resolved time-correlated single-photon counting (TCSPC).** The desired fundamental wavelengths (870, 930, 945, and 960 nm, 0.4–0.6 W, 100 fs pulse duration) were generated with a tunable (700–980 nm) Tsunami Ti-Sapphire oscillator (80 MHz, Spectra-physics), and were then subject to third harmonic generation with a commercial tripler (TPL fs tripler, minioptic) to obtain the corresponding UV excitation light (290, 310, 315, and 320 nm) with about 100-fs pulse duration. The power of the excitation beam was attenuated to around 0.4 μW before being directed to the sample chamber of a commercially available FluoTime 200 system (Picoquant), where the protein samples (about 50 μM) were placed in a 5-mm cuvette for measuring. The instrument response function (FWHM 40 ps, shown in Supplementary Fig. 15a) was determined by measuring the scattering signal of UVR8 lysis buffer (150 mM NaCl, 25 mM Tris, and pH = 8.0). With convoluted multiple-exponential decay fitting (Eq. (5)), the time resolution is about 15 ps. Limited by the 80 MHz pump laser repetition rate, the time window is about 12 ns.

**Sub-nanosecond-resolved time-correlated single-photon counting (TCSPC).** Sub-nanosecond resolved TCSPC data were acquired using the commercially available FluoTime 200 system (Picoquant) with PLS-290 pulsed LED (8 MHz, 290 nm, ~1 μW, Picoquant) as the excitation source. The FWHM of instrument response function (IRF) is about 700 ps (shown in Supplementary Fig. 15a). The time window is about 120 ns, covering the whole range of tryptophan fluorescence decay. All samples (10 μM) were kept in 5-mm quartz cuvettes during measurements.

Fluorescence decay transients at 305, 310, 315, 320, 325, 330, 335, 340, 345, 350, 355, 360, 365, 370, 375, and 380 nm (16 wavelengths) were taken for all samples (data shown in Supplementary Fig. 1) at a magic angle (54.7°). For WT and $6W_d + 4W_c$ samples, no observable dissociation was detected after the measurements, as confirmed using a Superdex200 size column.

**Construction of lifetime-associated spectra.**

$$S(t) = \int_{-\infty}^{+\infty} \text{IRF}(\tau) \sum_i A_i e^{-(t-\tau)/\tau_i} d\tau + A_{\text{scat}}\text{IRF}(t) \tag{5}$$

All sub-nanosecond-resolved TCSPC data (Supplementary Fig. 1) were fitted using FluoFit software (Picoquant) with a convoluted multiple-exponential decay

model described in Eq. (5), where $S(t)$ is the measured signal; $IRF(t)$ is the instrument response function as measured by buffer scattering signal (shown in Supplementary Fig. 15a); $A_i$ and $\tau_i$ are the amplitude and time constant of the $i$th exponential component; and $A_{scat}$ is the contribution of scattering to total signal. For each sample, transients of 16 different wavelengths were globally fitted with a same set of time constants. After the fitting parameters were obtained (shown in Supplementary Tables 6–14), with the following relationship between steady-state emission spectrum ($I_{ss}$) and exponential decay times and amplitudes:

$$I_{ss}(\lambda) = \int_0^{+\infty} \sum_i A_i e^{-\frac{t}{\tau_i}} dt = \sum_i A_i \tau_i, \tag{6}$$

the spectrum of the $m$th exponential component (as shown in Fig. 2b–f and Supplementary Fig. 2a–d) was decomposed from the total steady-state emission by the following equation:

$$I_m(\lambda) = I_{ss}(\lambda) \frac{A_m \tau_m}{\sum_i A_i \tau_i}, \tag{7}$$

where $I_m(\lambda)$ is the intensity of the lifetime-associated spectrum of the $m$th exponential component; $A_i$ and $\tau_i$ are the amplitude and time constant of the $i$th exponential component from the TCSPC transient at wavelength $\lambda$ ($\lambda = 305, 310, 315, 320, 325, 330, 335, 340, 345, 350, 355, 360, 365, 370, 375,$ and $380$ nm); $I_{ss}(\lambda)$ is the intensity of the steady-state emission spectrum of the same sample. The time constants of the fastest component (0.45 or 0.5 ns) were fixed in all samples. Here we define the 2nd lifetime of distal tryptophan in the presence of the acceptors as $\tau_{DA2}$. For WT and $6W_d + 4W_c$ only, constraints were applied to $\tau_{DA2}$ to allow the $\tau_{DA2}$ emission spectrum to be similar to $6W_d2$ (2.7 ns) spectrum (Supplementary Fig. 3). Otherwise, $\tau_{DA2}$ (1.2 or 1.5 ns) will mix with $\tau_3$ (1.4 ns), rendering both components not being accurately resolved.

**MD simulations**. The X-ray structure of UVR8 (PDB entry: 4D9S) was solvated in a cubic box of $104 \times 104 \times 104$ Å$^3$ using TIP3P water model. Counter ions were added to neutralize charge and produce an ion concentration of ~0.15 M. The simulation of the wild type was set up using CHARMM c38a2[38]. CHARMM27 force field[39] with CMAP correction[40] was used. The system was first subjected to energy minimization, followed by gradually heating from 10 to 300 K every 10 K using 100-ps NPT simulations at each temperature. During energy minimization and heating, harmonic restraints with the force constant of 5.0 kcal mol$^{-1}$ Å$^{-2}$) for backbone and 4.0 kcal mol$^{-1}$ Å$^{-2}$) for heavy atoms of sidechain were kept on the non-water and non-counter ions. The simulation was first carried out by using CHARMM during initial 20-ns equilibration. Then long-time simulation up to 2 μs using GROMACS-4.6.5 molecular dynamics code[41] at 300 K and 1 atm pressure with a 2-fs time step. The non-pair list was updated every 10 steps. The grid neighbor searching method was applied in the simulation with a 10 Å cutoff distance for the short-range neighbor list. Electrostatic interactions were treated by using the Partical–Mesh Ewald (PME) summation method[42] with a 14 Å for long range and 10 Å for short-range electrostatic cutoff, respectively. The short-range cutoff for van der Waals interactions during the simulation was 12 Å. The isotropic pressure coupling was achieved by Parrinello–Rahman method[43] with a compressibility of $4.5 \times 10^{-5}$ bar$^{-1}$.

**QM/MM calculation**. We extracted 500 snapshots (one snapshot every 4 ns) from the 2 μs production simulation for QM/MM calculations using a locally modified version of GAMESS code[44] in CHARMM quantum part. In QM/MM calculations, the electron singlet excited state can be written as[45]:

$$\Phi(S_1) = \underbrace{\Psi_1 \cdots \hat{A}\{\Psi_a^{S_1}\} \cdots \Psi_N}_{N-1\text{ fragments}}, \tag{8}$$

where $\hat{A}\{\Psi_a^{S_1}\}$ specifies an antisymmetric wave function for the locally excited residue $a$. It can be constructed using the corresponding fragment time-dependent KS orbitals. The other $N-1$ fragments are treated as classically molecular mechanics (MM). For every snapshot, the transition dipole moment was calculated for each of the 13 tryptophan chromophores using time-dependent range-separated hybrid functional *TD-CAM-B3LYP* with *6-31+G(d)* basis set. A total number of 6500 (13 × 500) of QM/MM calculations were conducted.

**Energy-transfer rate and time constant calculations for individual donor–acceptor pairs with FRET theory**. Details about FRET theory can be found elsewhere[28]. Briefly, resonance energy-transfer rates $k_{RET}$ (in ns$^{-1}$) were calculated as follows:

$$k_{RET} = \frac{1}{\tau_D}\left(\frac{R_0}{R}\right)^6 = 8.79 \times 10^{23} \frac{\kappa^2 Q_D J}{R^6 n^4 \tau_D}, \tag{9}$$

$J$ is the spectral overlap integral (in cm$^3$ M$^{-1}$) between donor emission spectrum and acceptor absorption spectrum (shown in Fig. 1f–h), which was obtained using Eq. (4).

$R$ (in Å) is the center-to-center distance between the donor and acceptor based on MD simulation trajectories or on X-ray structure[16,17]. We used the midpoint of the $C_{3a}$–$C_{7a}$ bond of indole ring as the center of the indole chromophores for

distance calculations. $\kappa^2$ is the orientation factor defined as follows:

$$\kappa^2 = (\mathbf{\mu_A} \cdot \mathbf{\mu_D} - 3(\mathbf{\mu_A} \cdot \mathbf{R})(\mathbf{\mu_D} \cdot \mathbf{R}))^2, \tag{10}$$

where $\mathbf{\mu_A}$ and $\mathbf{\mu_D}$ are the unit transition dipole moments of the acceptor and the donor, respectively, which were determined either by QM/MM calculations as described above or by X-ray structure[16,17] with previously measured transition dipole moment of indole $1L_a$ state[46]. $\mathbf{R}$ is the unit vector connecting centers of donor and acceptor tryptophan residues. The calculated $\kappa^2$ and distances based on static X-ray structure are shown in Supplementary Tables 15 and 16. $n$ is the refractive index within protein and the value of 1.33 was used[47]. $Q_D$ is the fluorescence quantum yield (QY) of donor without acceptors. $\tau_D$ is the donor lifetime in the units of nanosecond.

For distal tryptophan donors ($W_d$), the fluorescence QY of mono-$6W_d$ (0.15, Supplementary Fig. 15b) was used in calculations. Since tryptophan has two lifetimes, the amplitude weighted average lifetime was used here:

$$\tau_D = \frac{A_1 \tau_1 + A_2 \tau_2}{A_1 + A_2} = \frac{0.29 \times 0.5 + 0.71 \times 2.7}{0.29 + 0.71} = 2.06 \text{ ns} \tag{11}$$

In Eq. (11), $\tau_1$ and $\tau_2$ are 0.5 and 2.7 ns, as determined by experiments. $A_1$ and $A_2$ are the corresponding amplitudes, which are 0.29 and 0.71, respectively (Fig. 2a).

For peripheral tryptophan donors ($W_p$), the average fluorescence quantum yield measured with $6W_d + 3W_p$ (0.33, Supplementary Fig. 15c) was used for FRET calculations. The amplitude weighted average lifetime measured with $6W_d + 3W_p$ was calculated similarly:

$$\tau_D = \frac{A_1 \tau_1 + A_2 \tau_2}{A_1 + A_2} = \frac{0.11 \times 1.1 + 0.89 \times 7.0}{0.11 + 0.89} = 6.35 \text{ ns} \tag{12}$$

In Eq. (12), $\tau_1$ and $\tau_2$ are 1.1 and 7.0 ns, as determined with $6W_d + 3W_p$. $A_1$ and $A_2$ are the corresponding amplitudes, which are 0.11 and 0.89, respectively (Supplementary Fig. 7g).

For the static X-ray structure, corresponding energy-transfer timescales $\tau_{RET}$ (in ns) for each donor–acceptor pair (Supplementary Table 3) were calculated using Eq. (13):

$$\tau_{RET} = \frac{1}{k_{RET}} \tag{13}$$

To consider conformational distribution of UVR8, the RET rates in 500 snapshots from a 2-μs simulation trajectory were calculated. For every snapshot, the transition dipole moment was calculated for each tryptophan chromophore using described QM/MM methods above. Energy-transfer rates were calculated using Eq. (9). For each donor–acceptor pair, 500 RET rates were obtained from 500 MD structures. RET rate distributions were plotted in Fig. 3a–c and Supplementary Fig. 5.

For $W_d$ to $W_p$ and $W_c$ energy transfer, we simulated the excited state decay dynamics of the 2.7-ns component of every donor in the presence of individual acceptor with Eq. (14):

$$[W^*]_t = \frac{1}{500}\sum_{i=1}^{500} \exp(-(k_{RET,i} + 1/2.7)t) = \exp(-t/\tau_{DA}), \tag{14}$$

where $k_{RET,i}$ is the calculated RET rate (Eq. (9)) in the $i$th MD structure for each donor–acceptor pair, $t$ is the time. This numerical simulation was conducted for all 84 pairs of $W_d$ to $W_p/W_c$ (shown in Supplementary Fig. 6). All curves can be fitted well with single-exponential decay with various time constants ($\tau_{DA}$) as labeled in Supplementary Fig. 6. The effective RET time constant was obtained as:

$$\tau_{RET} = 1/(\tau_{DA}^{-1} - 2.7^{-1}) = 1/k_{RET} \tag{15}$$

The resulting energy-transfer timescales $\tau_{RET}$ (in ns) are shown in Supplementary Table 3.

For $W_p$ to $W_c$ energy transfer, we simulated the total energy-transfer dynamics to $4W_c$ as follows:

$$[W^*]_t =$$
$$\frac{1}{500}\sum_{i=1}^{500}\exp(-(k_{RET285,i} + k_{RET233,i} + k_{RET94,i} + k_{RET337,i} + k_{RET285b,i} + k_{RET233b,i} + k_{RET94b,i} + k_{RET337b,i})t),$$
$$\tag{16}$$

where $k_{RET285,i}$, $k_{RET233,i}$, $k_{RET94,i}$, $k_{RET337,i}$, $k_{RET285b,i}$, $k_{RET233b,i}$, $k_{RET94b,i}$, and $k_{RET337b,i}$ are calculated RET rates to W285, W233, W94, W337, W285b, W233b, W94b, and W337b, respectively, in $i$th MD snapshot. $[W^*]_t$ is the simulated excited state decay curve of certain $W_p$ due to energy transfer to $4W_c$. All three curves were fitted with double exponential decay model (as shown in Fig. 3d).

**Total RET rates of tryptophan donors**. Briefly, as shown in Fig. 2k, every $W_d$ transfers energy to all interfacial W residues on both UVR8 subunits. With RET rates of individual donor–acceptor pairs, total RET rates were obtained by summing all parallel RET rates on the same donor (as shown in Eq. (17)).

$$k_{total,Wm} = \frac{1}{\tau_{total,Wm}} = \sum_i^N k_{RET,WmWi} = \sum_i^N \frac{1}{\tau_{RET,WmWi}} \tag{17}$$

$k_{total,Wm}$ and $\tau_{total,Wm}$ are the total RET rate and time constant for $Wm$. $k_{RET,WmWi}$ is the effective RET rate (defined in Eq. (15)) from one distal tryptophan m ($Wm$,

$m = 39, 92, 144, 196, 300, 352$) to one interfacial tryptophan $i$ ($Wi$). $\tau_{\mathrm{RET},WmWi}$ is the corresponding time constant shown in Supplementary Table 3. $N$ is the number of interfacial tryptophan residues in the protein. Similarly, the $k_{\mathrm{total}}$ and $\tau_{\mathrm{total}}$ were obtained for energy transfer from $W_p$ to $W_c$ for each of the three $W_p$, using Eq. (17). However, energy transfer to the cluster that is further away was not considered ($N = 4$ in Eq. (17) for WT) due to negligible contributions to the total transfer rates (see Supplementary Table 17).

**Numerical simulations of the original 2.7 ns component decay dynamics**. Here we define the 2nd lifetime (original 2.7 ns) of distal tryptophan with the acceptors as $\tau_{\mathrm{DA2}}$. For each $W_d$ ($Wm$, $m = 39, 92, 144, 196, 300$, or 352), by adding the rates of the total energy transfer (from Eq. (17)) and the original 2.7 ns channel, the decay time constant can be obtained (Eq. (18)).

$$\tau_{\mathrm{DA2},Wm} = 1/(1/2.7 + 1/\tau_{\mathrm{total},Wm}) \tag{18}$$

By summing up the decay dynamics of every single $W_d$ (colored dashed lines in Fig. 2h–j and Supplementary Fig. 2e–j), we simulated the total fluorescence dynamics (black solid lines in Fig. 2h–j and Supplementary Fig. 2e–j) based on theoretical RET calculations:

$$\mathrm{Simul.}(t) = \sum_m^{W_d} \exp(-t/\tau_{\mathrm{DA2},Wm}) \tag{19}$$

The overall decay dynamics of $6W_d$ is the sum of the 6 decay curves of individual $W_d$.

**Numerical simulations of picosecond resolved TCSPC transients using kinetic models**. The procedure is detailed in the Supplementary Methods. Briefly, with the energy-transfer model shown in Figs. 2k and 3e and experimentally measured fluorescence lifetimes of various tryptophan, we first simulated the excited state population evolution of three W groups with kinetic models described in the Supplementary Methods. After convoluted with the instrument response function, signal contributions from all tryptophan groups were added together to give the total simulation curve. The associated spectra of each group, as shown in Fig. 3l–o, were constructed by decomposing the total steady-state emission spectra of each mutant based on the time integration of simulation curve of three tryptophan groups.

**Energy-transfer efficiency calculations**. Details about energy-transfer efficiency calculations using energy-transfer rates can be found elsewhere[28]. We treat two lifetime components of tryptophan as two subpopulations, whose ratios are the amplitude ratios from fitting of fluorescence decays. As above mentioned, distal tryptophan residues have two lifetimes: $\tau_1$ and $\tau_2$, which are 0.5 and 2.7 ns. To consider the overall RET efficiency of the two different subpopulations, we used the population weighted average efficiency for each distal tryptophan $Wm$ ($m = 39, 92, 144, 196, 300$, or 352):

$$E_{Wm} = (1 - R_{D2})E_1 + R_{D2}E_2 \tag{20}$$

in which, $R_{D2}$ is the amplitude ratio of the 2.7 ns lifetime, 0.71; $E_1$ and $E_2$ are the RET efficiencies of the two lifetime components and were calculated as follows:

$$E_1 = 1 - \frac{(0.5^{-1} + \tau_{\mathrm{total},Wm}^{-1})^{-1}}{0.5} \tag{21}$$

$$E_2 = 1 - \frac{(2.7^{-1} + \tau_{\mathrm{total},Wm}^{-1})^{-1}}{2.7} \tag{22}$$

The overall RET efficiency for all $6W_d$ is the arithmetic average of the RET efficiencies of 6 individual $W_d$ calculated from Eq. (20). All results are shown in Supplementary Table 4.

$$E = \frac{\sum E_{Wm}}{6} \tag{23}$$

The energy-transfer efficiencies from W198, W250, and W302 to the pyramid center were calculated with four subpopulations, which are based on their fluorescence lifetimes and RET rates. Similarly, the population weighted average efficiency for each $W_p$ is:

$$E_{Wi} = (1 - R_{\mathrm{slow},Wi})\left[(1 - R_{2,Wi})\left(1 - \frac{(\tau_{1,Wi}^{-1} + \tau_{\mathrm{total1},Wi}^{-1})^{-1}}{\tau_{1,Wi}}\right) + R_{2,Wi}\left(1 - \frac{(\tau_{2,Wi}^{-1} + \tau_{\mathrm{total1},Wi}^{-1})^{-1}}{\tau_{2,Wi}}\right)\right]$$

$$+ R_{\mathrm{slow},Wi}\left[(1 - R_{2,Wi})\left(1 - \frac{(\tau_{1,Wi}^{-1} + \tau_{\mathrm{total2},Wi}^{-1})^{-1}}{\tau_{1,Wi}}\right) + R_{2,Wi}\left(1 - \frac{(\tau_{2,Wi}^{-1} + \tau_{\mathrm{total2},Wi}^{-1})^{-1}}{\tau_{2,Wi}}\right)\right] \tag{24}$$

where $R_{2,Wi}$ is the amplitude ratio of the slower lifetime of $Wi$ ($i = 198, 250$, or 302); $\tau_{1,Wi}$ and $\tau_{2,Wi}$ are the faster (1–2 ns) and slower (6–8 ns) lifetimes of $Wi$, respectively. $\tau_{\mathrm{total1},Wi}$ and $\tau_{\mathrm{total2},Wi}$ are the fast and slow total RET timescales of $Wi$ as shown in Fig. 3d. $R_{\mathrm{slow},Wi}$ is the percentage of the slow energy transfer. All values are shown in Supplementary Table 18. Overall light-perception efficiency by UVR8 was calculated with energy-transfer efficiency of individual tryptophan and absorbance values of three tryptophan groups (Supplementary Table 5) at 290 nm. Detailed procedure is in the Supplementary Methods.

## Data availability

The authors declare that the data supporting the findings of this study are available within the paper and its Supplementary Information files.

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

## Acknowledgements
We thank Prof. Yigong Shi (Tsinghua University) for generously providing the UVR8 plasmid, and Prof. Maria-Elisabeth Michel-Beyerle (Nanyang Technological University) for stimulating discussions. This work was supported in part by the National Institute of Health (Grant GM118332 to D.Z. for experiments and GM46736 to J.G. for computation) and the National Natural Science Foundation of China (for support of collaboration effort through a visit of X.L. and a sabbatical stay of D.Z. in Shanghai Jiao Tong University and Grant No. 21533003 to J.G. for support of H.R. to complete computational work).

## Author contributions
D.Z. designed the research. X.L., M.K., Z.L., F.Z., and L.W. performed the experiments. H.R. and J.G. did computational studies. X.L. and D.Z. wrote the paper. All authors discussed and edited the paper.

## Competing interests
The authors declare no competing interests.
