## [Peer Review File · Nature Communications]

Reviewers' Comments:

Reviewer #1:

Remarks to the Author:

UVR8 is a 7-bladed beta-propeller protein that functions as a UV-B photoreceptor in its homodimeric form. Each UVR8 monomer contains 14 tryptophans: 6 distal (Wd), 3 peripheral (Wp), 4 central (Wc; pyramid), and 1 in the C-terminus. The present understanding is that in particular W285, together with W233, (both belonging to the group of 4 central tryptophans) plays a key role for UV-B responsiveness of UVR8. Li et al. used ultrafast fluorescence spectroscopy and quantum computational calculations to describe the relative contributions of all three studied tryptophan groups (Wp, Wd, Wc) to UV-B absorption and energy transfer to the key Wc residues (W285 and W233). This is an interesting and extensive study that provides an important step forward towards understanding how UVR8 works as a UV-B photoreceptor. The authors provide evidence that light-perception quantum efficiency increases from 35% to 73% through the additional UV-B photon harvesting and energy transfer through the Wd and Wp tryptophans.

I only have a few points that should be addressed:

- The 14th Trp in the C-terminus should at least be mentioned. Otherwise the text makes the impression UVR8 contains only 13 Trps.
- Consider to add at least 1-2 sentences in the introduction (line 56) what happens after monomerization of UVR8 in its signaling pathway (e.g. interaction and inhibition of the E3 ubiquitin ligase COP1).
- In table S1 W302H is indicated – mention as is done for R286A* (mono-6Wd)
- In the discussion it would be interesting to shortly mention and discuss the evolutionary conservation of the studied tryptophan residues in plant UVR8 proteins (e.g. including comparison with *Chlamydomonas* UVR8 that has been functionally studied in quite some detail, next to *Arabidopsis* UVR8). It seems that the architecture of tryptophan residues and thus likely also the UV-B activation mechanism has arisen early on in the evolution of UVR8 photoreceptors (derived from a RCC1-type protein).
- A recent study suggested that UVR8 also responds to UV-A under natural conditions (Rai et al., *Plant Cell Environ* 2020); it may be interesting to shortly discuss.

Reviewer #2:

Remarks to the Author:

The use of naturally-occurring amino acids such as tryptophan as UV sensors is not particularly novel and has been well documented for at least ten years. However, the assembly of 26 tryptophan residues into a complex antenna system with an energy gradient, as described here, to increase the efficiency of UV perception is intriguing. By a combination of time resolved fluorescence measurements, mutational analysis, and mixed quantum-classical simulations the authors present a convincing analysis of energy transfer pathways to the "reaction center" pyramid that initiates monomerization.

However, I felt the paper was written in such away that its appeal to abroad audience would be limited. To me, the paper was too focused on the details of the fluorescence decay analysis to the detriment of the bigger picture - exactly how the excited tryptophans initiate monomerization is left to the reader to find out. What consequences and opportunities enhanced sensing of UV light brings to plants is not mentioned. Both the Introduction and the Discussion seemed very Terse and lacking context. I suspect this will lessen the impact of the paper.

The energy transfer is described exclusively as localized hopping by the Forster mechanism. What are the couplings between the closest trypts? Are all the levels confined to a single chromophore? Do the spectral shifts giving the directionality of the energy flow arise only from environmental effects or also from trypt-trypt interactions?

Reviewer #3:

Remarks to the Author:

A work entitled "A leap in quantum efficiency through light harvesting in photoreceptor UVR8" has been presented by Xiankun Li, Haisheng Ren, Mainak Kundu, Zheyun Liu, Frank W. Zhong, Lijuan Wang, Jiali Gao and Dongping Zhong. Using a combined experimental and theoretical approach they study the energy-transfer network involving 26 tryptophan residues in the UVR8 receptor. Quantitative agreement between time-resolved fluorescence data and QM/MM-based results has been found, for what concerns the total resonant-energy-transfer time constants and the related energy-transfer pathways. Furthermore, the tryptophan residues have been grouped into three different sets, assigning a specific role to them in the energy transfer.

As a general comment, I found the main conclusions convincing, and the applied strategy reliable and robust. The present work deserves to be published on Nature Communications, but, in my opinion, a number of changes should be applied to make the manuscript clearer and accessible: Results, Discussion and Methods Sections should be reshaped in a more consistent way, to provide a precise information when needed, as specified below. My comments are given as follows:

- 1) The authors should enrich the Methods section with, at least, a part of the Supporting Information, especially that regarding the energy-transfer and population dynamics: this modeling is an essential part of the work, and should be included in the main text;
- 2) In many points of Results (lines 123-126, 131-134, 142-144, 159-162, 186-190, 198-199), the authors report numbers of energy-transfer times, transfer efficiency etc. with no reference to the method and/or formulas used to compute them: formulas are in Supporting Information, but a reference is necessary to avoid the reader loses the focus. For this reason, I suggest, as already mentioned, to move part of the first ten pages of the Supporting Information into Methods and to explicitly refer to the applied formulas. The last four subsections of Methods ("Energy transfer...", "Numerical...", "Model..." and "Energy transfer efficiency...") are currently too general and do not provide any specific insight into the employed physical models;
- 3) Did the authors computationally study the reason why the mono-6Wd mutant is a monomer?
- 4) In which conditions (temperature) the spectra have been recorded?
- 5) In Figs 2b-f, spectra of 6Wd1 and 6Wd2 mutants are reported: the two species have been not defined in the main text; only a short comment in the caption of Figure 2 is given, which refers generically to the Supporting Information;
- 6) in order to save clarity, author should specify the meaning of all the numbers on lines 131-134;
- 7) the authors should briefly comment the choice, as an example, of W302 instead of the other residues in Figure 2b; if no physical reason is behind such a choice, a sentence should be however added;
- 8) Computed energy-transfer rates in Figures 2g and 3 a-d show a different probability distributions; are these differences due to the relative mobility of the involved pairs? Did the authors explore this behaviour?

9) the "ab" subscript in the formula on line 327 is not defined.

10) Can an excitonic model, as typically used for light-harvesting complexes (e.g., LH2 and FMO), be applied to study the tryptophan network at quantum level?

Few typos:

i) "were" instead of "was" on line 246;

ii) "GAMESS" instead of "GAMSS" on line 324;

iii) "RET" and "FRET" are used to indicate the same group of quantities.

Responses to review reports

Reviewer 1

UVR8 is a 7-bladed beta-propeller protein that functions as a UV-B photoreceptor in its homodimeric form. Each UVR8 monomer contains 14 tryptophans: 6 distal (W_d), 3 peripheral (W_p), 4 central (W_c; pyramid), and 1 in the C-terminus. The present understanding is that in particular W285, together with W233, (both belonging to the group of 4 central tryptophans) plays a key role for UV-B responsiveness of UVR8. Li et al. used ultrafast fluorescence spectroscopy and quantum computational calculations to describe the relative contributions of all three studied tryptophan groups (W_p, W_d, W_c) to UV-B absorption and energy transfer to the key W_c residues (W285 and W233). This is an interesting and extensive study that provides an important step forward towards understanding how UVR8 works as a UV-B photoreceptor. The authors provide evidence that light-perception quantum efficiency increases from 35% to 73% through the additional UV-B photon harvesting and energy transfer through the W_d and W_p tryptophans.

Respond: We are very glad that the reviewer 1 is so positive on our work with great enthusiasm. As the reviewer said, this study revealed a huge increase of light-perception quantum efficiency from 35% to 73% through light harvesting, providing important insight into the molecular mechanism of photoreceptor UVR8.

I only have a few points that should be addressed:

- The 14th Trp in the C-terminus should at least be mentioned. Otherwise the text makes the impression UVR8 contains only 13 Trps.

Respond: The reviewer is right, and each monomer indeed has 14 tryptophan residues. But the 14th Trp (W400) is located at an unstructured C-terminus, which may not be essential to UVR8 function as shown by mutation work, probably not significant for the light harvesting reported here. UVR8 in other species, such as *Chlamydomonas* UVR8, even does not have such a C-terminal Trp. To clarify, we changed the sentence in line 59 to “Each UVR8 monomer has 14 tryptophan residues, and except the unstructured C-terminal one, the rest 13 structural Trp could be classified into 3 distinct groups, a distal ring (6W_d in Fig. 1b), a peripheral outlier (3W_p in Fig. 1c) and a pyramid center (4W_c in Fig. 1c).”

- Consider to add at least 1-2 sentences in the introduction (line 56) what happens after monomerization of UVR8 in its signaling pathway (e.g. interaction and inhibition of the E3 ubiquitin ligase COP1).

Respond: This is an excellent suggestion. In the introduction, we added “Followed by initial light perception, monomeric UVR8 interacts with signaling partners including

E3 ubiquitin-protein ligase COP1 (CONSTITUTIVE PHOTOMORPHOGENIC1) and accumulates in the nucleus, regulating expression of various downstream genes.”

- In table S1 W302H is indicated – mention as is done for R286A (mono-6Wd)*

Respond: In our mutant design and purification, we finally found W94/233/285/337/198/250F/W302H/R286A mutant has a higher protein yield than the phenylalanine mutant W94/233/285/337/198/250F/W302F/R286A. We have added one more note under Table S1 to explain the reason of using histidine instead of phenylalanine to replace W302.

- In the discussion it would be interesting to shortly mention and discuss the evolutionary conservation of the studied tryptophan residues in plant UVR8 proteins (e.g. including comparison with Chlamydomonas UVR8 that has been functionally studied in quite some detail, next to Arabidopsis UVR8). It seems that the architecture of tryptophan residues and thus likely also the UV-B activation mechanism has arisen early on in the evolution of UVR8 photoreceptors (derived from a RCC1-type protein).

Respond: The reviewer provided a very good perspective. Other studies compared UVR8 sequences from Arabidopsis and other species. Chlamydomonas UVR8 (CrUVR8) has 12 tryptophan residues, lacking W144 (one W_d) and W400 (C-terminal Trp) in AtUVR8. Chlorophyceae and Trebouxiophyceae UVR8 have all 14 Trp in AtUVR8. It is likely that the light harvesting as well as the Trp pyramid in UVR8 is conserved among the all UVR8 from the evolution of the RCC1 family. It will be interesting to investigate the different light harvesting efficiency and subsequent activation in the pyramid. As suggested by the reviewer, we added a short discussion about the possible conserved light-harvesting mechanism.

- A recent study suggested that UVR8 also responds to UV-A under natural conditions (Rai et al., Plant Cell Environ 2020); it may be interesting to shortly discuss.

Respond: Thank the reviewer for pointing out this recent reference. From the above-mentioned paper, UV-A light from 320 to 335 nm can also trigger UVR8 monomerization. Under natural conditions, UVR8 can respond to part of UV-A light. This finding is very important and is consistent with our results. From our data, 4W_c absorption extends to 330-335 nm (Fig. 2d and 2e), allowing UVR8 to absorb part of UV-A region. However, the region of 340-350 nm is too red to be absorbed even by 4W_c. Thus, we have added one sentence “The red-side absorption of 4W_c explains the recently reported UVR8 monomerization in response to UV-A light below 335 nm wavelength.”

Reviewer 2

The use of naturally-occurring amino acids such as tryptophan as UV sensors is not particularly novel and has been well documented for at least ten years. However, the assembly of 26 tryptophan residues into a complex antenna system with an energy gradient, as described here, to increase the efficiency of UV perception is intriguing. By a combination of time resolved fluorescence measurements, mutational analysis, and mixed quantum-classical simulations the authors present a convincing analysis of energy transfer pathways to the “reaction center” pyramid that initiates monomerization.

Respond: We thank the reviewer 2 for his/her positive comments with great enthusiasm. As the reviewer kindly reminded, the use of aromatic amino acids for UV sensing may not be novel and has been reported in artificial systems. However, in biology, UVR8 is the only known UV photoreceptor/sensor in nature. Other proteins/peptides also absorb UV light with aromatic amino acids and may have excitation energy transfer processes that may not be relevant to biological functions. Photosynthesis has light harvesting systems using prosthetic pigments, not amino acids. To the best of our knowledge, UVR8 is the only system in nature that uses natural amino acids for both light harvesting and light perception functions.

However, I felt the paper was written in such a way that its appeal to abroad audience would be limited. To me, the paper was too focused on the details of the fluorescence decay analysis to the detriment of the bigger picture - exactly how the excited tryptophans initiate monomerization is left to the reader to find out. What consequences and opportunities enhanced sensing of UV light brings to plants is not mentioned. Both the Introduction and the Discussion seemed very Terse and lacking context. I suspect this will lessen the impact of the paper.

Respond: We thank the reviewer 2 for these constructive suggestions. We focused on data analyses to facilitate readers to follow our study procedure. The dissociation mechanism after excitation of 4W_c remains highly controversial and our detailed studies are on the way. As suggested by the reviewer, we added more discussion to show readers a broader picture.

The energy transfer is described exclusively as localized hopping by the Forster mechanism. What are the couplings between the closest trypts? Are all the levels confined to a single chromophore? Do the spectral shifts giving the directionality of the energy flow arise only from environmental effects or also from trypt-trypt interactions?

Respond: The reviewer 2 asked an excellent question for the coupling between adjacent Trp residues and it will be addressed in detail in the future with theoretical calculations. For RET rate calculations, the dipole-dipole coupling is a good

approximation due to long donor-acceptor distances. This is absolutely true for the case of W_d to W_p/W_c energy-transfer pairs since the distances are generally above 15 Å. For W_p to W_c , although the couplings beyond dipole-dipole interactions may exist, Forster mechanism also agrees reasonably well with the experimental data.

The evidence of exciton coupling in $4W_c$ has been observed by previous/current circular dichroism (CD) spectra. From our mutation studies, it seems W285 and W233 in $4W_c$ are critical and without either of the two, the exciton signature of CD spectra disappear but without either the other two of W94 and W337 the exciton character still remains. But such exciton coupling was not observed in W_d and W_p or between W_p and W_c , suggesting W_d and W_p excited states are well localized. Thus, the spectral tuning of W_d and W_p only comes from environments. The absorption of $4W_c$ extending to the red side of 320 nm is possibly the evidence of exciton coupling among W_c . From our study, the red emission of $4W_c$ is mainly due to a nearby negatively residue D129 (aspartic acid 129). We do not know how strong the exciton coupling and there are no high-level theoretical calculations yet. We are collaborating with Prof. Jiali Gao in University of Minnesota (one of the authors) and performing such calculations. Such exciton coupling could delocalize excited states in the pyramid, but mainly relating to UVR8 dimer dissociation. Overall, we also do not believe the coupling is very strong because there is no evidence of delocalized charge-transfer character in the ground state and we only observed a small red-tail weak absorption to 320 nm. Thus, on the current stage, our analyses probably provide the best quantitative picture.

Reviewer 3

A work entitled “A leap in quantum efficiency through light harvesting in photoreceptor UVR8” has been presented by Xiankun Li, Haisheng Ren, Mainak Kundu, Zheyun Liu, Frank W. Zhong, Lijuan Wang, Jiali Gao and Dongping Zhong. Using a combined experimental and theoretical approach they study the energy-transfer network involving 26 tryptophan residues in the UVR8 receptor. Quantitative agreement between time-resolved fluorescence data and QM/MM-based results has been found, for what concerns the total resonant-energy-transfer time constants and the related energy-transfer pathways. Furthermore, the tryptophan residues have been grouped into three different sets, assigning a specific role to them in the energy transfer.

As a general comment, I found the main conclusions convincing, and the applied strategy reliable and robust. The present work deserves to be published on Nature Communications, but, in my opinion, a number of changes should be applied to make the manuscript clearer and accessible: Results, Discussion and Methods Sections should be reshaped in a more consistent way, to provide a precise information when needed, as specified below. My comments are given as follows:

Respond: We thank the reviewer 3 for the positive comments on our work with great enthusiasm. We totally agree with the reviewer that the work deserves to be published on Nature Communications. We carefully address the concerns the reviewer pointed out below.

1) The authors should enrich the Methods section with, at least, a part of the Supporting Information, especially that regarding the energy-transfer and population dynamics: this modeling is an essential part of the work, and should be included in the main text;

Respond: The reviewer raised a good point. We have restructured our Method section by moving FRET theory calculations, total RET rates and certain numerical simulation methods from the Supplementary Information to the main text in the Method section. Hopefully, this change will make readers easier to follow our study procedure.

2) In many points of Results (lines 123-126, 131-134, 142-144, 159-162, 186-190, 198-199), the authors report numbers of energy-transfer times, transfer efficiency etc. with no reference to the method and/or formulas used to compute them: formulas are in Supporting Information, but a reference is necessary to avoid the reader loose the focus. For this reason, I suggest, as already mentioned, to move part of the first ten pages of the Supporting Information into Methods and to explicitly refer to the applied formulas. The last four subsections of Methods (“Energy transfer...”, “Numerical...”, “Model...”

and “Energy transfer efficiency...” are currently too general and do not provide any specific insight into the employed physical models;

Respond: These are very good suggestions. We followed the suggestion and also have numbered the equations and cited equation numbers whenever necessary.

3) Did the authors computationally study the reason why the mono-6Wd mutant is a monomer?

Respond: The reviewer raised a great scientific question. R286 forms critical salt-bridges with negatively charged D96/D107 from the other monomer. Previous mutagenesis studies have shown the mutation of R286 to any neutral residues destroys this critical electrostatic interaction and even the single mutant R286A is monomeric. We are collaborating with Prof. Jiali Gao in University of Minnesota (one of the authors) and performing MD calculations after neutralizing the charge on R286. The whole UVR8 dissociation process takes milliseconds, which is beyond our simulation time window. But in an initial 2 microseconds trajectory, we observed the salt-bridge network at the interface is disrupted.

4) In which conditions (temperature) the spectra have been recorded?

Respond: All measurements were done at room temperature. We followed the reviewer’s suggestion and added temperature information in the Methods part.

5) In Figs 2b-f, spectra of 6Wd1 and 6Wd2 mutants are reported: the two species have been not defined in the main text; only a short comment in the caption of Figure 2 is given, which refers generically to the Supporting Information;

Respond: 6W_{d1} and 6W_{d2} are the two W_d lifetimes 0.5 ns and 2.7 ns, respectively. We decomposed their associated spectra by fitting TCSPC data of Mono-6W_d (See Supplementary Figure 3). With amplitude percentages at various wavelengths (Supplementary Figure 3b) and equation 7, we readily obtained the spectra of the 0.5-ns (6W_{d1}) and 2.7-ns (6W_{d2}) components.

6) in order to save clarity, author should specify the meaning of all the numbers on lines 131-134;

Respond: In those lines, we wrote “Thus, we determined total energy-transfer efficiency of 45% from the interior 6W_d to the interfacial tryptophan residues 3W_p and 4W_c. The two additional times of 1.4 and 5.4 ns for 6W_d+4W_c are from 4W_c”

(Supplementary Fig. 4) but the 1.4 and 5.5 ns for the WT are from $4W_c$ and $3W_p$, respectively. For the former, the resulting emission spectrum (hexagons in Fig. 2e) is the same as directly obtained by excitation at 315 nm (solid line). Numbers in those lines are all experimentally measured fluorescence lifetimes in various mutants.”

45% is the overall RET efficiency of the $6W_d$ in WT, as calculated using the measured 1.2-ns lifetime in WT. This 1.2-ns component is changed from the original 2.7 ns ($6W_d$ in Mono- $6W_d$) due to RET channels in WT. Thus, RET time should be $1/(1/1.2-1/2.7)$ ns or 2.16 ns. Plugging into equations 20-22, the efficiency was calculated as 45%. However, this is just an estimate by assuming all $6W_d$ have a same RET time of 2.16 ns. In reality, $6W_d$ have different RET time scales and thus different efficiency values (supplementary table 4). The overall efficiency should be the average of the six numbers (equation 23 and supplementary table 4). Later with the whole network solved, the obtained number is 0.44 (supplementary table 4, overall efficiency $6W_d$ to $3W_p+4W_c$) and agrees very well with the 45% estimate.

The 1.4 and 5.4 ns are experimentally measured time scales in the mutant of $6W_d+4W_c$. They are both from $4W_c$. 1.4 ns is consistent with 315-nm excitation data in Figure 3g (the second lifetime of $4W_c$). 5.4 ns is caused by W302 mutation as shown in Supplementary Figure 4. Without W302 mutation, $4W_c$ only has 80 ps and 1.4 ns lifetimes.

The 1.4-ns and 5.5-ns components were observed in WT. 1.4 ns is from $4W_c$; 5.5-ns is from slow RET population of $3W_p$.

7) the authors should briefly comment the choice, as an example, of W302 instead of the other residues in Figure 2b; if no physical reason is behind such a choice, a sentence should be however added;

Respond: In Figure 2b, we intended to show only one of the three $6W_d+1W_p$ data to avoid the redundancy. We chose $6W_d+W302$ instead of $6W_d+W250$ or $6W_d+W198$ with no scientific reason.

8) Computed energy-transfer rates in Figures 2g and 3a-d show a different probability distributions; are these differences due to the relative mobility of the involved pairs? Did the authors explore this behaviour?

Respond: Figure 2g shows rate distributions for RET from the $6W_d$ to interfacial $3W_p$ and $4W_c$; Figure 3a-d gives rate distributions from $3W_p$ to $4W_c$. We agree with the reviewer that the distributions depend on relative mobility of the pairs, which is another interesting question. Both distance (R) fluctuations and relative rotations affect κ^2/R^6 and thus the rate distributions. Generally, RET rates are more sensitive to R fluctuations

for shorter donor-acceptor distances. For example, a fluctuation from 2 Å to 3 Å brings much larger change to RET rates than a fluctuation from 10 Å to 11 Å. That explains why closer pairs have broader probability distributions. To further elucidate the origin of different rate distributions, a careful look at tryptophan motions is needed. Unfortunately, we have not done any in-depth analysis of tryptophan local motions at the current stage. With the MD trajectories, we will explore the local mobility of each tryptophan residues in the future.

9) the “ab” subscript in the formula on line 327 is not defined.

Respond: We thank the reviewer for the kind reminding. It is a typo. We removed the “ab” subscript in the equation in response to the reviewer’s concern.

10) Can an excitonic model, as typically used for light-harvesting complexes (e.g., LH2 and FMO), be applied to study the tryptophan network at quantum level?

Respond: The reviewer raised a good question here. The exciton coupling between tryptophans can be experimentally observed by circular dichroism (CD) spectra. From our studies and previous studies, we only found the exciton coupling signature among the 4W_c. But such exciton coupling was not observed among 6W_d and 3W_p or between different Trp groups. It seems exciton coupling can only exist in very short ranges between the closely packed 4W_c. An excitonic model may be applied to study energy transfer within the 4W_c in the future but may not be necessary on the current stage.

Few typos:

i) “were” instead of “was” on line 246;

ii) “GAMESS” instead of “GAMSS” on line 324;

iii) “RET” and “FRET” are used to indicate the same group of quantities.

Respond: We thank the reviewer for his/her careful reading. We have fixed the typos accordingly.

Reviewers' Comments:

Reviewer #2:

Remarks to the Author:

The authors have addressed my concerns and comments and in my view the paper is now appropriate for publication.

Graham Fleming

Response to the reviewer 2

We are happy that the reviewer 2 is satisfied with our response and no more response is needed.